# Dileucine ingestion, but not leucine, increases lower body strength and performance following resistance training: A double-blind, randomized, placebo-controlled trial

**Anthony M. Hagele[1], Joesi M. Krieger[1], Connor J. Gaige[1], Kevin F. Holley[1], Kristen N. Gross[1], Joshua M. Iannotti[1], Leah E. Allen[1], Paige J. Sutton[1], Logan S. Orr[1], Petey W. Mumford[1], Martin Purpura[2,3], Ralf Jager[2,3], Chad M. Kerksick[1]***

**1** Exercise and Performance Nutrition Laboratory, Kinesiology Department, College of Science, Technology and Health, Lindenwood University, St. Charles, Missouri, United States of America, **2** Increnovo, LLC, Whitefish Bay, Wisconsin, United States of America, **3** Ingenious Ingredients L.P., Lewisville, Texas, United States of America

\* ckerksick@lindenwood.edu

**Data Availability Statement:** All relevant data are within the manuscript and its Supporting information Files.

## Abstract

### Background

The essential amino acid leucine (LEU) plays a crucial role in promoting resistance-training adaptations. Dileucine (DILEU), a LEU-LEU dipeptide, increases MPS rates, however its impact on resistance training outcomes remains unexplored. This study assessed the effects of DILEU supplementation on resistance training adaptations.

### Methods

Using a randomized, double-blind, placebo-controlled approach, 34 resistance-trained males (age: 28.3 ± 5.9 years) consumed 2 grams of either DILEU monohydrate (RAMPS™, Ingenious Ingredients, L.P.), LEU, or placebo (PLA) while following a 4-day per week resistance training program for 10 weeks. Changes in body composition, 1-repetition maximum (1RM) and repetitions to failure (RTF) for leg press (LP) and bench press (BP), anaerobic capacity, countermovement jump (CMJ), and maximal voluntary contraction (MVC) were assessed after 0 and 10 weeks.

### Results

Significant main effects for time ($p < 0.001$) were realized for LP and BP 1RM and RTF. A significant group × time interaction was identified for changes in LP 1RM ($p = 0.02$) and LP RTF ($p = 0.03$). Greater increases in LP 1RM were observed in DILEU compared to PLA ($p = 0.02$; 95% CI: 5.8, 73.2 kg), and greater increases in LP RTF in DILEU compared to LEU ($p = 0.04$; 95% CI: 0.58, 20.3 reps). No significant differences were found in other measures.

**Funding:** Funding was acquired by CMK. Grant #: 01-2020. This study was funded by Ingenious Ingredients, L.P. (https://ing2.com/). Martin Purpura and Ralf Jager are principals of Ingenious Ingredients, L.P., the sponsor of the study. The funders assisted in conceptualizing and designing the study, and reviewing and editing the manuscript. The funders were not involved in data collection, data analysis, or data interpretation. Additionally, the sponsors played no role in the decision to publish, prepare, or revising the manuscript.

**Competing interests:** I have read the journal's policy and the authors of this manuscript have the following competing interests: M.P and R.J are principals of Ingenious Ingredients, L.P., the sponsor of the study, and inventors of numerous patent applications for the use of dileucine but have not been involved in data collection or analysis. All other authors declare no competing interests. This does not alter our adherence to PLOS ONE policies on sharing data and materials as there are no restrictions on sharing data and/or materials.

## Conclusions

DILEU supplementation at 2 grams daily enhanced lower body strength and muscular endurance in resistance-trained males more effectively than LEU or PLA. These findings suggest DILEU as a potentially effective supplement for improving adaptations to resistance training.

NCT06121869 retrospectively registered.

## Introduction

Skeletal muscle, known for its remarkable adaptability to external stimuli such as exercise, and nutrition [1], plays a critical role in the context of athletic performance [2]. The building and maintenance of muscle mass, which directly influences muscle strength, is effectively achieved through resistance training using an appropriate stimulus [3]. Such training induces an increase in muscle protein synthesis (MPS) via mechanical load [4], which can be amplified through adequate nutrient availability [5].

Over the last decade, leucine (LEU), an essential amino acid, has garnered considerable attention for its role in stimulating MPS in both animal and human models [6, 7]. As an anabolic activator for the mammalian target of rapamycin complex 1 (mTORC1) pathway, LEU facilitates the assembly of MPS machinery at the ribosomal level [8]. The mTORC1 pathway integrates signals from nutrient availability and resistance exercise, enhancing MPS through the phosphorylation of key proteins such as the ribosomal protein S6 kinase (p70S6K) and the eukaryotic translation initiation factor 4E-binding protein (4E-BP1) [9, 10]. This process underscores LEU's potential in amplifying the anabolic response when timed with exercise.

Achieving a certain LEU threshold within skeletal muscle is critical for maximizing the pathway's activation. Empirical evidence suggests a dose-dependent relationship for LEU, with 1.7 to 3.9 g being optimal for stimulating MPS and facilitating muscle recovery post-exercise [11–13]. The LEU content in various protein sources varies significantly, with whey protein containing the highest content (~12–14%), and animal (8–9% for non-dairy sources) and dairy (>10%) generally having higher amounts than plant proteins (6–8%) [14, 15]. This variation affects their anabolic potential, as proteins with higher LEU content are likely to more effectively trigger the mTORC1 pathway, thus promoting greater muscle recovery and growth [16, 17]. Chronic intake of LEU, especially when aligned with a resistance training regimen, has been shown to enhance muscle strength and hypertrophy, leading to exploration of LEU-rich supplements to enhance adaptations to resistance training [18].

The focus within nutritional science has shifted towards understanding the role of food-borne peptides (e.g., di- or tripeptides) and their incorporation as components of anabolic feeding formulations. Early evidence has suggested absorption rates of dipeptides can be similar to, and at times even exceed those of comparable amino acids [19]. Dileucine (DILEU), a peptide comprised of two LEU molecules, represents a particularly intriguing compound due to its unique dipeptide structure, and its potential to facilitate more efficient absorption and utilization than LEU alone [20]. DILEU is transported into the intestinal endothelium via the PepT1 H$^+$/peptide cotransporter, potentially making it more rapidly available to muscle tissue, expediting and amplifying its anabolic effects post-ingestion [21]. Despite its promising profile, research into the direct impact of DILEU supplementation remains sparse. Foundational evidence by Paulussen et al. [20] demonstrated that DILEU elevates plasma concentrations and stimulates MPS rates more effectively than LEU alone. Additionally, the rate at which

LEU concentrations rise in the bloodstream could be a determining factor for the anabolic efficiency of dietary proteins [22]. As Paulussen et al. [20] showed, DILEU absorption rates and area under the curve values were significantly elevated, suggesting that DILEU supplementation could create a more potent or prolonged anabolic environment conducive to muscle growth and recovery. However empirical evidence to support this theory is notably lacking.

The primary aim of this study was to compare the observed changes in resistance-training adaptations after supplementation of similar amounts of LEU, DILEU, or a placebo (PLA) in healthy resistance-trained males following a 10-week resistance training program. The choice of a 10-week duration for this study is supported by previous research showing that significant changes in resistance training outcomes can be observed over a period of 8–12 weeks [23–25]. Given the role of LEU in protein synthesis and its documented dose-response relationship with MPS [26], it was plausible that DILEU could offer a more pronounced benefit due to its dipeptide structure. This speculation is grounded in the understanding that peptides can exert different physiological effects compared to their constituent amino acids when ingested in isolation [27]. We hypothesize that DILEU supplementation would result in greater resistance training adaptations, resulting in improvements in muscular strength, muscular endurance, and power along with improvements in body composition when compared to LEU or a PLA for 10-weeks while engaged in a heavy resistance training program.

## Methods

### Experimental design

The study utilized a randomized, double-blind approach where participants were equally distributed into groups based on their fat-free mass to ensure a balanced representation in each group. The study was conducted according to the guidelines of the Declaration of Helsinki and approved by the Institutional Review Board of Lindenwood University (IRB-21-64; Date: 18 DEC 2020). Recruitment for the study began 7 MAY 2021 and ended 29 JUN 2023. Written informed consent was obtained from all participants prior to their participation in the study. A priori statistical analysis of previous strength outcomes using a similar study design [25] using G*Power [28] revealed that achieving an effect size of 0.25–0.50 with an alpha level of 0.05 and power of 0.80, using a $3 \times 2$ mixed factorial ANOVA with repeated measures on time, would require a sample size of 8–11 per group for a total of 24–33 participants. Participants supplemented with 2 g of either DILEU, LEU, or PLA for 10-weeks. The supplementation period coincided with a linear periodized, resistance training program, consisting of two upper body and two lower body workouts each week, totaling 40 workouts. Supplements were ingested within 60-minutes post-workout on training days, or with the first meal of the day on non-training days. Resting assessments included body mass, body water (total, extracellular, intracellular), body composition (fat mass, fat-free mass, dry lean mass, and % fat), and muscle thickness (mid-thigh and vastus lateralis). Performance assessments included muscular strength (one-repetition maximum [1RM]) and muscular endurance (repetitions to fatigue at a load of 80% 1RM, [RTF]) were evaluated using the bench press and leg press exercise. Additionally, we conducted countermovement jumps to assess lower-body power, isometric mid-thigh pull to assess total body force production, and Wingate anaerobic capacity tests to assess anaerobic power. All anthropometric, body composition, and performance assessments were completed during an initial screening visit, and repeated after 0, 2, 6, and 10 of weeks of resistance training. Before each laboratory visit, participants fasted for 8 hours and refrained from exercise, caffeine, nicotine, and alcohol for a minimum of 24 hours to ensure accurate and consistent measurement conditions. Participants were provided nutritional recommendations to ensure adequate energy ($>30$ g·kg$^{-1}$·d$^{-1}$) and protein consumption ($>1.5$ g·kg$^{-1}$·d$^{-1}$) aiming

to facilitate positive training adaptations and reduce the potential influence of differing dietary intakes [29, 30]. A compliance threshold of 90% was set, with participants falling below this level being subject to removal from the study. This study was retroactively registered on clinicaltrials.gov (NCT06121869).

## Study participants

Healthy, resistance trained males ($n$ = 34, age 28 ± 6 years, height: 176.0 ± 7.4 cm, weight: 78.4 ± 10.9 kg, body mass index: 25.5 ± 3.7 kg·m$^{-2}$, body fat %: 19.1 ± 3.9% fat) completed the entire study protocol. To be eligible, participants self-reported at least 1 year of resistance training experience, could bench press ≥ 1.0x their body weight, could leg press ≥ 1.5x their body weight, and had a BMI less than 25 kg·m$^{-2}$. Participants with a BMI ≥ 25 were accepted if their body fat % (determined by DXA) was ≤ 25% fat. Additionally, participants were required to discontinue all ergogenic nutritional supplements (e.g., creatine monohydrate, β-alanine) except for multi-vitamins/minerals for 30 days before and during participation in the study. Participants currently using or reporting usage of anabolic-androgenic steroids within the past 12 months were excluded.

## Anthropometric assessments

During the initial assessment, participants' height was measured using a wall-mounted stadiometer (HR-200, Tanita Corp, Inc, Tokyo, Japan) to the nearest ±0.5 cm, without shoes. Body mass was measured on each study visit with a self-calibrating digital scale (Tanita BWB-627A, Tokyo, Japan) to the nearest ±0.1 kg. Weight stability was confirmed by comparing screening visit and week 0 body masses; a deviation of more than 2% was deemed non-weight stable and resulted in exclusion from the study.

## Body composition

Body composition was assessed using dual-energy X-ray absorptiometry (DXA) with the Hologic QDR Horizon W system (Hologic, Inc., Bedford, MA, USA). A trained research team member conducted and analyzed all scans. The DXA device was calibrated daily in accordance with the manufacturer's recommendations, and data analysis was conducted using the provided software (Hologic APEX Software, Version 5.6, Hologic Inc., Bedford, MA, USA). Before all body composition assessments, participants provided a urine sample (2–5 mL) for analysis of urine-specific gravity (USG) using a handheld refractometer (Aichose, XSC Co., Ltd., Guangdong, CHN). Participants with a USG value ≤ 1.020 were considered well-hydrated, with participants above this USG threshold were provided water ad libitum until their USG reached ≤ 1.020 or rescheduled to a different day. Due to a critical failure of the DXA machine that required replacement, DXA variables were computed with 6 finishers in the LEU group, 7 finishers in the DILEU group, and 9 finishers in the PLA group. Measured body composition parameters included fat-free mass, fat mass, and % body fat. The test-retest reliability of DXA measurements for fat mass (CV: 3.9%, ICC: 0.96) and fat-free mass (CV: 1.14%, ICC: 0.99).

Following DXA assessments, total body water (TBW), intracellular body water, and extracellular body water was assessed using bioelectrical impedance analysis (BIA) (InBody 570, InBody, Beverly Hills, CA, USA). Test-retest reliability using our device has been previously established for BIA TBW (CV: 7.9%, ICC: 0.99).

After completing the BIA assessment, mid-thigh (MT) and vastus lateralis (VL) muscle thickness were measured using a GE Doppler Ultrasound Scanner (General Electric Healthcare, Chicago, IL, USA) equipped with a multi-frequency linear-array transducer (Logiq S7 R2

Expert, General Electric Healthcare, Chicago, IL, USA). MT images were captured at the mid-way point between the mid-inguinal crease and the lateral epicondyle of the femur in the transverse plane, while VL images were taken at the mid-point of the VL between the greater trochanter and the lateral epicondyle of the femur. Both MT and VL measures were taken after participants rested in a supine position for 10 minutes. All images were collected at a depth in which the edge of the femur was visible, and this depth was held constant for all image collection timepoints. All ultrasound settings (frequency: 10 MHz, gain: 50db, dynamic range: 75), with the exception of depth, were held constant across all participants and time points. Three images per participant were captured at each timepoint and averaged. Following the conclusion of the study, images were analyzed using ImageJ software (Version 1.54g, National Institutes of Health, Bethesda, MD, USA). MT and VL thickness was measured using the straight-line function and defined as the distance between the subcutaneous adipose tissue-vastus lateralis interface and the deep aponeurosis. To establish measurement reliability, the same experienced rater performed all measurements for each participant. Test-retest reliability was previously determined for MT (CV 4.08%, ICC 0.96) and VL (CV: 1.03%, ICC: 0.98).

## Performance assessments

**Maximal strength.** Muscular strength was assessed through 1RM measurements using both the leg press and bench press exercises. Before determining 1RM, participants completed a standardized lower- and upper-body warm-up consisting of simple stretches and body weight movements. Using a protocol consistent with the recommendations of the National Strength and Conditioning Association [31], participants completed one set of 10 repetitions using only the sled (for leg press) or barbell (for bench press). The warm-up progressed in a systematic manner including five repetitions at 50% of their perceived 1RM, three repetitions at 75% of their perceived 1RM, and 90% of their perceived 1RM. A two-minute rest was observed between each set. One-repetition sets were then completed with progressively increasing loads until 1RM was determined within three to five one-repetition attempts, with two minutes of rest between each attempt. Subsequent 1RM assessments (weeks 2, 6, and 10) were completed using the participant's previously established 1RM as a reference for establishing loads during the testing. Participants had a minimum of five minutes of rest between determination of their 1RM and completion of the next test. All leg press repetitions were performed on a commercial, 45-degree leg press machine (XFW-7800 Leg Press, True Fitness, St. Louis, MO, USA). Foot position and hip angle were standardized by recording the heel position during the initial testing and maintain this position for future 1RM determinations. Participants initiated each leg press repetition from the bottom position, with their knees in approximately 90 degrees of flexion, and concentrically contracted their legs to fully extend the knees while completing each repetition. They were required to keep their hands clear of their knees, thighs, and torso. All bench press repetitions were completed using a standard adjustable bench press and knurled barbell (Rogue 20kg Ohio Power Bar, Rogue Fitness, Columbus, OH, USA). Hand spacing was standardized for each set by recording the width of the hands. Adhering to technique standards, participants were required to maintain five points of contact during all bench press repetitions and lower the bar to the sternum and press back until both elbows reached full extension. Two experienced research team members were present to ensure proper technique for both exercises. Total strength was computed by calculating the sum of leg press 1RM and bench press 1RM for each individual.

**Muscular endurance.** Approximately five minutes after establishing the respective 1RM for leg press and bench press, participants completed lower- and upper-body RTF assessments using a load corresponding to 80% of their week 0 1RM for both leg press and bench press,

respectively. Participants performed as many repetitions as possible until failure, while maintaining proper lifting technique and full range of motion throughout all repetitions. Each test was terminated when technique failure occurred throughout any repetition or if the participant paused for more than two seconds between repetitions. The number of successfully completed repetitions was counted and recorded. A five-minute rest was observed before proceeding to the next test. These tests were conducted in the exercise lab and were supervised by trained research assistants. Total repetitions was computed by calculating the sum of leg press repetitions and bench press repetitions completed for each individual.

**Countermovement jump.** Bilateral countermovement jumps were completed to evaluate lower-body power production. Participants performed five maximal jumps on force platforms sampling at 1000Hz (Hawkins Dynamics, Westbrook, ME, USA) with a 30-second rest between each jump. Participants began each jump in an athletic stance with their hands on their hips, then performed an initial downward movement by flexing at the knees and hips, which were then immediately extended for a maximal vertical jump. Hands remained on hips throughout the entire repetition. The trial with the highest jump height was recorded.

**Isometric mid-thigh pull test.** Maximal force production was assessed using an isometric mid-thigh pull (IMTP). Participants performed three five-second maximal pulls, with a minute of rest between each repetition. Participants were positioned according to the methodology of Comfort et al. [32] using a custom rig apparatus with two uniaxial force plates sampling at 1000Hz (PASCO Scientific, Roseville, CA, USA). An adjustable-height horizontal bar was attached to the mid-thigh pull rig, with the bar height recorded during the initial assessment and replicated during all testing sessions. Participants were verbally encouraged to pull upward as hard and fast as possible using a double overhand grip and wrist wraps. The highest peak force of the three attempts was recorded.

**Anaerobic capacity.** Anaerobic capacity was assessed using the Wingate anaerobic capacity test on a magnetically braked cycle ergometer (Lode Excalibur Sport, Groningen, NED). The resistance for all Wingate testing was set at 7.5% [33] of their Week 0 body weight (kg) for each participant and was not changed for any subsequent testing. The testing protocol began with a 60-second warm-up consisting of light pedaling ($\leq$ 90 rpm) against zero resistance. After the warm-up, participants were provided with a five-second count down where they were instructed to increase their pedaling cadence to reach their maximum cadence. At the end of the five-second count down and with each participant pedaling at their maximum cadence, the resistance was applied by the cycle ergometer and participants were instructed to continue pedaling as fast as possible against their allotted resistance for the 30-second test. Verbal encouragement was provided throughout the 30-second sprint, with no feedback regarding elapsed time. Saddle height and position, and handlebar height and depth were recorded during the initial assessment and standardized for each subsequent test. Peak power, average power, total work, and fatigue index were computed and used as indicators of anaerobic power.

### Resistance training program

A template of the resistance training program is outlined in Table 1. Participants were provided with paper training cards following completion of the week 0 performance assessment and updated their training log after each workout. The resistance training program followed a linear, split-body periodization program with two upper-body and two lower-body workouts each week [23]. A progressive overload scheme was followed to facilitate increases in strength and muscle mass. For the first six weeks (weeks 1–6), each workout consisted of three sets of ten repetitions at a 10-repetition max (RM) load. On the final set of each exercise, participants

**Table 1. Sample resistance training program.**

| Weeks | Day 1, Day 3 | Day 2, Day 4 |
|---|---|---|
| 1–6[a] | Bench press, 3 × 10 RM | Back squat or leg press, 3 × 10 RM |
| | Chest flies, 3 × 10 RM | Leg extension, 3 × 10 RM |
| | Lat pulldown, 3 × 10 RM | Romanian deadlift, 3 × 10 RM |
| | Seated row, 3 × 10 RM | Split squat, 3 × 10 RM |
| | Shoulder press, 3 × 10 RM | Leg curl, 3 × 10 RM |
| | Shoulder shrug, 3 × 10 RM | Calf raise, 3 × 10 RM |
| | Biceps curl, 3 × 10 RM | Ab crunches, 3 × 25 |
| | Triceps extension, 3 × 10 RM | |
| 7–10[b] | Bench press, 4 × 6 RM | Back squat or leg press, 4 × 6 RM |
| | Chest flies, 4 × 6 RM | Leg extension, 4 × 6 RM |
| | Lat pulldown, 4 × 6 RM | Romanian deadlift, 4 × 6 RM |
| | Seated row, 4 × 6 RM | Split squat, 4 × 6 RM |
| | Shoulder press, 4 × 6 RM | Leg curl, 4 × 6 RM |
| | Shoulder shrug, 4 × 6 RM | Calf raise, 4 × 6 RM |
| | Biceps curl, 4 × 6 RM | Ab crunches, 3 × 25 |
| | Triceps extension, 4 × 6 RM | |

[a] One-minute rest between sets

[b] Two-minutes rest between sets

performed as many repetitions as they were able. Following the autoregulatory model introduced by Mann et al. [34], if participants were able to complete 12 or more repetitions on their final set, they were instructed to increase the load for their next workout. During the final four weeks (weeks 7–10), each workout consisted of four sets of six repetitions at a 6 RM. Again, participants completed as many repetitions as they were able to on their final set. If participants completed seven or more repetitions on their final set, they were assigned to the next highest load for their next workout [34]. One minute of rest was allotted between sets for weeks 1–6, while two minutes of rest were allotted between sets for weeks 7–10. Each resistance training session took approximately 60 minutes to complete. Completion of the program was not directly supervised. To maximize ecological validity, participants completed their workouts in the facility of their choosing, provided they had access to all equipment necessary to complete the exercises within the program.

## Dietary protocol

After the week 0 performance assessment, participants were provided daily dietary recommendations. A range of daily caloric needs was estimated for each study participant by calculating resting energy expenditure using an average of the Harris-Benedict [35] and Mifflin-St. Joer [36] formulas and then multiplying that value by an activity factor of 1.6 and 1.8. Participants were also instructed to maintain a daily protein intake of 1.6 to 1.8 g of protein per kilogram of body mass [29]. Participants were required to log their dietary intake at least three days before each study visit using an online dietary assessment tool (ASA-24; https://asa24.nci.nih.gov/; Accessed: 6 Dec 2023). To achieve compliance with the dietary recommendations outlined above, study participants were provided a binder that outlined their recommended energy and protein intakes throughout the study protocol. Participants were given sample meal plans with recommended meal options to meet their goals, and examples of how to successfully complete

the food recall in addition to graphic-based examples of portion size estimators. Food recall records were reviewed by laboratory staff during each study visit to assess whether participants met energy and protein requirements throughout the study.

## Supplementation protocol

Following performance assessments at week 0 and before beginning the resistance-training program, participants were randomly assigned in a double-blind fashion based on their baseline DXA fat-free mass using an online software program (https://randomizer.org; Accessed 6 Dec 2023). Participants were asked to ingest isomolar amounts of either 2 g of DILEU monohydrate (L-Leucyl-L-Leucine monohydrate as RAMPS™, Ingenious Ingredients, L.P., Lewisville, TX, USA), 2 g of LEU (NNB Nutrition, Nanjing, China), or 2 g of PLA (resistant starch, NNB Nutrition, Nanjing, China) daily, and were required to return to the laboratory every 30 days to receive additional supplement. Each dose was consumed in capsule form and ingested along with eight ounces of water. On workout days, participants ingested their assigned dose within 60 minutes of completing their workout. On non-workout days, one dose was ingested with their morning meal. Additional analytical verification was completed which revealed the test product to be 99.7% L-Leucyl-L-Leucine monohydrate.

## Compliance monitoring

Participants maintained a participant diary in their workout binder, logging their supplement consumption on both training and non-training days. Compliance with the supplementation regimen was monitored when participants returned to the laboratory to receive an additional 30 doses of their assigned supplement. During these visits, research team members reviewed capsule counts and participant diaries and confirmed any questions with the study protocol. Supplement compliance was calculated as the percentage of days in which compliance was achieved divided by the number of days in the protocol.

To monitor compliance with the resistance training program, participants were instructed to complete paper-based training cards, recording exercise choice, reps, and loads. Research team members monitored the completion of the logs weekly and at each study visit, using email or phone calls to facilitate compliance. Participants were also required to submit a photograph of themselves at the gym, documenting their presence before or after workouts. Compliance was calculated as the percentage of completed workouts.

## Adverse event reporting

The occurrence of adverse events was recorded throughout the entire duration of the study using spontaneous reporting by the study participants, interaction of a research team member with a study participant, or through review of a study participant's research file. All recorded events were systematically categorized using MedDRA system organ class and lowest level terms (LLT) before being graded using Common Terminology Criteria for Adverse Events ([CTCAE] Version 5.0, U.S. Department of Health, and Human Services (published: November 27, 2017)).

## Statistical analysis

All statistical analysis was completed in a blinded fashion using IBM SPSS 27 (Armonk, NY, USA), with figures generated using GraphPad (La Jolla, CA, USA). Data are presented as means ± standard deviations. For all dependent measures, descriptive statistics (means and standard deviations) were calculated. Data was first analyzed for normality, skewness, and

kurtosis. All non-normal data was log-transformed prior to analysis. For all statistical tests, data was considered statistically significant when the probability of type 1 error was 0.095 or less. The primary endpoints of this analysis were considered to be the delta (Week 10 –Week 0) value for DXA fat-free mass and leg press 1RM. Secondary endpoints were the delta (Week 10 –Week 0) values for DXA fat mass, DXA lean mass, and DXA % body fat, along with bench press 1RM, bench press RTF, leg press RTF, total strength, total repetitions, peak force from IMTP, jump height, peak propulsive force, peak anaerobic power, mean anaerobic power, and total work. A $3 \times 2$ mixed factorial (group × time) ANOVA with repeated measures on time were used to determine any statistically significant differences for time and group main effects and group × time interactions. Further, delta changes were calculated, and between-group differences of these changes were evaluated using one-way ANOVA with Tukey post-hoc tests being applied. Additionally, 95% confidence intervals were constructed on the between-group differences of the observed changes from baseline.

## Results

### Subject compliance and baseline characteristics

The Consolidated Standards of Reporting Trials (CONSORT) diagram for this study is presented in Fig 1. Briefly, a total of 587 potential participants were recruited for the study. Of these individuals, 141 failed pre-screening, and 113 were consented. Of the 113 participants that provided consent and began the study, 25 declined to participate and 32 did not qualify. Of the 57 that qualified and began the study, 34 successfully completed the intervention (PLA $n = 12$, LEU $n = 11$, DILEU $n = 11$).

There were no baseline differences between supplement groups for select dependent variables related to age, body composition, or strength (see Table 2 for $p$-values). Overall, supplement and resistance training compliance was 97.6%.

### Self-reported dietary intakes

Self-reported dietary data from Week 0 and Week 10 were analyzed using raw dietary intake and dietary intake that was normalized to each participant's body mass. As seen in Table 3, group × time interactions, time, and group effects for all non-normalized data for energy (kcal·d$^{-1}$) (group × time: $p = 0.63$; group: $p = 0.34$; time: $p = 0.26$), carbohydrate (g·d$^{-1}$) (group × time: $p = 0.53$; group: $p = 0.18$; time: $p = 0.21$), protein (g·d$^{-1}$) (group × time: $p = 0.71$; group: $p = 0.50$; time: $p = 0.51$), and fat (g·d$^{-1}$) (group × time: $p = 0.34$; group: $p = 0.62$; time: $p = 0.35$) were non-significant. Additionally, similar outcomes were revealed when all data was represented relative to each person's recorded body mass: normalized energy (kcal·kg$^{-1}$·d$^{-1}$) (group × time: $p = 0.48$; group: $p = 0.47$; time: $p = 0.39$), normalized carbohydrate (g·kg$^{-1}$·d$^{-1}$) (group × time: $p = 0.62$; group: $p = 0.28$; time: $p = 0.24$), normalized protein (g·kg$^{-1}$·d$^{-1}$) (group × time: $p = 0.79$; group: $p = 0.43$; time: $p = 0.75$), and normalized fat (g·kg$^{-1}$·d$^{-1}$) (group × time: $p = 0.23$; group: $p = 0.85$; time: $p = 0.51$).

### Changes in body mass, body water, and body composition

Table 4 presents the main effects of group, time, and group × time interaction on body mass, body water, and body composition variables. There were no significant main effects of time or group × time interactions for changes in fat mass, or % fat. However, significant main effects of time were observed for increases in body mass ($p < 0.001$), fat-free mass ($p = 0.005$), dry lean mass ($p < 0.001$), TBW ($p < 0.001$), ICW ($p < 0.001$), ECW ($p < 0.001$), and MT muscle thickness ($p = 0.04$). No significant changes were observed in VL muscle thickness ($p = 0.12$).

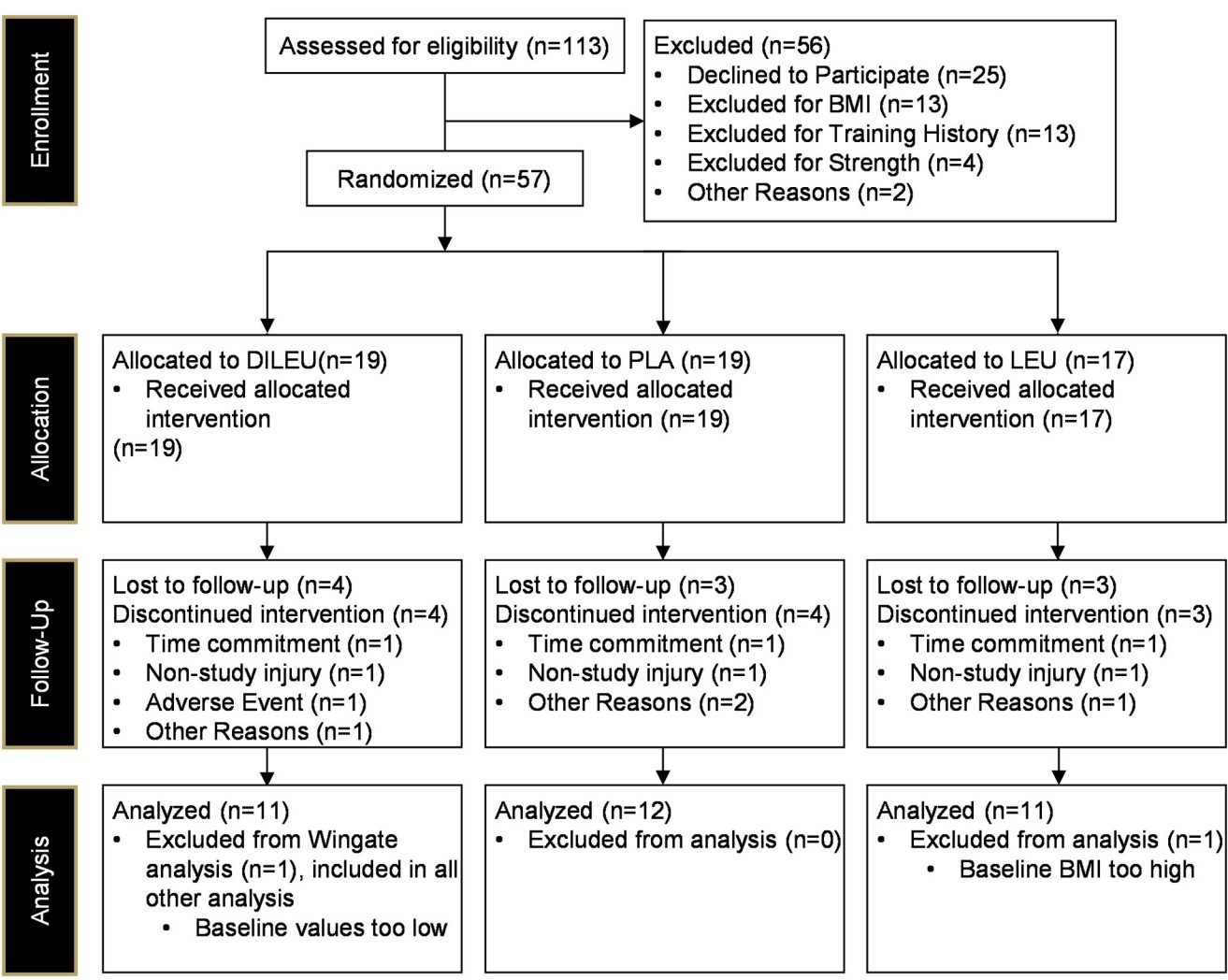

**Fig 1. CONSORT diagram.**

## Exercise performance

**Muscular strength and muscular endurance.** As seen in Table 5, changes in leg press 1RM (Fig 2) revealed a significant group × time interaction ($p = 0.02$). Pairwise comparisons revealed significant increase in leg press 1RM for DILEU compared to PLA ($p = 0.02$; 95% CI: 5.8, 73.2 kg). Changes in bench press 1RM indicated a significant main effect for time ($p < 0.001$), but no significant group × time interaction ($p = 0.16$) or main effect for group ($p = 0.46$). Changes in total strength indicated a significant group × time effect ($p = 0.02$). Pairwise comparisons revealed a significant increase in total strength for DILEU compared to PLA ($p = 0.02$; 95% CI: 6.8, 75.9 kg).

Changes in leg press RTF (Fig 3) indicated a significant group × time interaction ($p = 0.03$). Pairwise comparisons revealed significant increase in leg press RTF for DILEU compared to LEU ($p = 0.04$; 95% CI: 0.58, 20.3 reps). Changes in bench press RTF indicated a significant main effect for time ($p < 0.001$), but no significant group × time interaction ($p = 0.10$) or main effect for group ($p = 0.29$). Changes in total repetitions indicated a significant group × time

**Table 2. Baseline characteristics.**

| Variable | Group | Mean ± SD | p-value |
|---|---|---|---|
| Age (years) | LEU (n = 11) | 26.0 ± 6.3 | 0.17 |
| | DILEU (n = 11) | 30.7 ± 5.4 | |
| | PLA (n = 12) | 28.2 ± 5.4 | |
| Height (cm) | LEU | 175.5 ± 8.4 | 0.74 |
| | DILEU | 177.5 ± 7.7 | |
| | PLA | 175.1 ± 6.5 | |
| Body Mass (kg) | LEU | 81.2 ± 10.2 | 0.51 |
| | DILEU | 75.5 ± 8.2 | |
| | PLA | 78.3 ± 13.4 | |
| Body Mass Index (kg·m$^{-2}$) | LEU | 26.5 ± 4.1 | 0.41 |
| | DILEU | 24.3 ± 2.6 | |
| | PLA | 25.5 ± 4.0 | |
| DXA % Fat (%) | LEU (n = 6) | 17.8 ± 4.7 | 0.50 |
| | DILEU (n = 7) | 19.4 ± 3.6 | |
| | PLA (n = 9) | 19.7 ± 2.5 | |
| DXA Fat-Free Mass (kg) | LEU | 62.1 ± 8.9 | 0.48 |
| | DILEU | 58.5 ± 6.0 | |
| | PLA | 62.8 ± 7.0 | |
| Relative Leg Press (kg·kg$^{-1}$ $_{Body\ Mass}$) | LEU | 3.57 ± 0.62 | 0.72 |
| | DILEU | 3.37 ± 0.94 | |
| | PLA | 3.60 ± 0.56 | |
| Relative Bench Press (kg·kg$^{-1}$ $_{Body\ Mass}$) | LEU | 1.27 ± 0.23 | 0.53 |
| | DILEU | 1.16 ± 0.32 | |
| | PLA | 1.22 ± 0.15 | |

Data are presented as means ± SD; p = probability level of making Type I error; cm = centimeters; kg = kilograms; m = meters; DXA = dual energy x-ray absorptiometry; DILEU = dileucine; LEU = leucine; PLA = placebo

effect ($p < 0.01$), significant main effect for time ($p < 0.001$), and a significant group effect ($p = 0.04$). Pairwise comparisons revealed significant increase in total repetitions for DILEU compared to LEU ($p = 0.009$; 95% CI: 3.3, 25.3).

**Anaerobic capacity and performance metrics.** No significant group × time interactions were identified for relative peak power, relative mean power, total work, or fatigue index as assessed in the Wingate anaerobic capacity test. Changes in peak propulsive force during countermovement jump revealed a significant main effect for time ($p = 0.009$) but no significant group × time interaction ($p = 0.54$) while changes in jump height revealed no significant main effects of time ($p = 0.65$) or group × time interaction ($p = 0.92$). Changes in isometric mid-thigh pull performance revealed a significant main effect for time ($p = < 0.001$) but no group × time effect ($p = 0.13$) (Table 5).

## Discussion

This is the first study to examine the effects of daily supplementation with 2 g DILEU in resistance-trained males during a 10-week resistance training program on changes in resistance-training adaptations. The primary findings revealed a significant increase in leg press 1RM with DILEU supplementation in comparison to PLA. Moreover, DILEU supplementation

**Table 3. Dietary variables.**

| Variable | Group | Week 0 | Week 10 | Mixed Factorial (p) | |
|---|---|---|---|---|---|
| Energy (kcal·d$^{-1}$) | LEU | 2611 ± 839 | 2869 ± 790 | G | 0.34 |
| | DILEU | 2379 ± 682 | 2351 ± 732 | T | 0.26 |
| | PLA | 2583 ± 563 | 2801 ± 845 | G × T | 0.63 |
| Carbohydrate (g·d$^{-1}$) | LEU | 269.5 ± 86.1 | 304.6 ± 94.0 | G | 0.18 |
| | DILEU | 209.4 ± 66.4 | 232.1 ± 83.8 | T | 0.21 |
| | PLA | 279.7 ± 112.1 | 276.2 ± 130.0 | G × T | 0.53 |
| Protein (g·d$^{-1}$) | LEU | 129.2 ± 52.6 | 140.9 ± 52.7 | G | 0.50 |
| | DILEU | 121.4 ± 31.7 | 127.1 ± 47.7 | T | 0.51 |
| | PLA | 144.2 ± 33.2 | 141.2 ± 34.6 | G × T | 0.71 |
| Fat (g·d$^{-1}$) | LEU | 114.6 ± 48.5 | 123.2 ± 41.9 | G | 0.62 |
| | DILEU | 110.6 ± 42.6 | 104.1 ± 40.3 | T | 0.35 |
| | PLA | 97.1 ± 32.8 | 112.1 ± 37.2 | G × T | 0.34 |
| Relative Energy Intake (kcal·kg$^{-1}$·d$^{-1}$) | LEU | 32.4 ± 11.1 | 35.4 ± 11.5 | G | 0.47 |
| | DILEU | 31.0 ± 9.6 | 29.6 ± 9.1 | T | 0.39 |
| | PLA | 33.7 ± 8.8 | 36.8 ± 14.7 | G × T | 0.48 |
| Relative Carbohydrate (g·kg$^{-1}$·d$^{-1}$) | LEU | 3.35 ± 1.15 | 3.78 ± 1.41 | G | 0.28 |
| | DILEU | 2.71 ± 0.94 | 2.94 ± 1.18 | T | 0.24 |
| | PLA | 3.69 ± 1.67 | 3.68 ± 2.07 | G × T | 0.62 |
| Relative Protein (g·kg$^{-1}$·d$^{-1}$) | LEU | 1.58 ± 0.60 | 1.70 ± 0.66 | G | 0.43 |
| | DILEU | 1.59 ± 0.47 | 1.61 ± 0.56 | T | 0.75 |
| | PLA | 1.88 ± 0.54 | 1.84 ± 0.63 | G × T | 0.79 |
| Relative Fat (g·kg$^{-1}$·d$^{-1}$) | LEU | 1.43 ± 0.66 | 1.53 ± 0.60 | G | 0.85 |
| | DILEU | 1.45 ± 0.56 | 1.30 ± 0.47 | T | 0.51 |
| | PLA | 1.26 ± 0.45 | 1.48 ± 0.66 | G × T | 0.23 |

All variables relative to body mass use body mass obtained during Week 0. G × T = Interaction effect; T = Main effect for time; G = Main effect for group; $p$ = probability level of making Type I error; Kcal = kilocalories; g = grams; kg = kilograms; DILEU = dileucine; LEU = leucine; PLA = placebo; G × T = group × time

resulted in a greater increase in RTF compared to LEU supplementation. These results are significant in light of previous research [18], which has been largely inconclusive regarding the benefits of essential amino acid and LEU supplementation on resistance training adaptations.

The relationship between MPS and muscle protein breakdown following resistance training is a critical area of research. It is well-established that an acute bout of resistance training induces increases in both MPS and muscle protein breakdown [37, 38]. Notably and in the absence of feeding, the increase in muscle protein breakdown typically surpasses that of MPS, leading to a net negative muscle protein balance. This imbalance can be counteracted by the ingestion of essential amino acids, particularly in the range of 8–12 g, which enhance MPS and shift the balance towards net muscle protein accretion [39]. Among these amino acids, LEU is vital in stimulating MPS, acting as an "anabolic trigger" for mTORC1 related signaling [39] and the subsequent stimulation of postprandial MPS rates [40]. The relationship between plasma LEU concentrations and MPS follows a dose-dependent pattern, with ingestion of ~2.5 g LEU stimulating MPS to near maximal levels [26]. However, existing evidence indicates that when dietary protein intake is sufficient, additional free-amino acid or free-LEU supplementation does not significantly improve training outcomes [18].

Given this, the use of dipeptides like DILEU as an alternative becomes increasingly relevant. Dipeptides have been shown to be absorbed faster [41] and more efficiently than single free

**Table 4. Changes in body composition.**

| Variable | Group | Week 0 | Week 10 | Mixed Factorial (p) | | ES ($\eta^2$) | Pairwise Comparisons | | |
|---|---|---|---|---|---|---|---|---|---|
| | | | | | | | | 95% CI | (p) |
| Body Mass (kg) | LEU | 81.2 ± 10.2 | 82.7 ± 10.2 | G | 0.55 | 0.046 | LEU vs. DILEU | (-2.97, 1.25) | 0.58 |
| | DILEU | 75.5 ± 8.2 | 77.8 ± 8.9 | T | <0.001 | | LEU vs. PLA | (-1.71, 2.32) | 0.93 |
| | PLA | 78.3 ± 13.4 | 79.6 ± 13.2 | G × T | 0.49 | | DILEU vs. PLA | (-0.90, 3.23) | 0.36 |
| DXA Fat Mass (kg) | LEU | 14.0 ± 5.3 | 14.1 ± 5.1 | G | 0.61 | 0.051 | LEU vs. PLA | (-1.95, 1.55) | 0.96 |
| | DILEU | 14.8 ± 2.9 | 15.6 ± 3.9 | T | 0.17 | | DILEU vs. PLA | (-1.17, 2.17) | 0.73 |
| | PLA | 16.5 ± 4.3 | 16.8 ± 4.4 | G × T | 0.51 | | DILEU vs. LEU | (-1.15, 2.55) | 0.61 |
| DXA Fat-Free Mass (kg) | LEU | 62.1 ± 8.9 | 63.2 ± 9.0 | G | 0.50 | 0.015 | LEU vs. PLA | (-3.51, 2.36) | 0.87 |
| | DILEU | 58.5 ± 6.0 | 60.1 ± 6.8 | T | 0.005 | | DILEU vs. PLA | (-2.88, 2.73) | 0.99 |
| | PLA | 62.8 ± 7.2 | 64.6 ± 7.7 | G × T | 0.87 | | DILEU vs. LEU | (-2.59, 3.60) | 0.91 |
| DXA % Fat (%) | LEU | 17.6 ± 4.7 | 17.3 ± 4.7 | G | 0.40 | 0.073 | LEU vs. PLA | (-2.67, 1.23) | 0.63 |
| | DILEU | 19.4 ± 3.6 | 20.1 ± 4.5 | T | 0.40 | | DILEU vs. PLA | (-1.63, 2.10) | 0.94 |
| | PLA | 19.7 ± 2.5 | 20.2 ± 2.6 | G × T | 0.49 | | DILEU vs. LEU | (-1.10, 3.01) | 0.48 |
| Dry Lean Mass (kg) | LEU | 18.0 ± 2.4 | 18.3 ± 2.4 | G | 0.55 | 0.014 | LEU vs. PLA | (-0.41, 0.46) | 0.99 |
| | DILEU | 17.1 ± 2.0 | 17.4 ± 2.1 | T | <0.001 | | DILEU vs. PLA | (-0.33, 0.56) | 0.81 |
| | PLA | 16.7 ± 3.9 | 17.0 ± 3.9 | G × T | 0.81 | | DILEU vs. LEU | (-0.36, 0.54) | 0.88 |
| Total Body Water (L) | LEU | 49.0 ± 6.2 | 49.7 ± 6.1 | G | 0.58 | 0.008 | LEU vs. PLA | (-1.12, 0.85) | 0.94 |
| | DILEU | 46.4 ± 5.2 | 47.4 ± 5.3 | T | <0.001 | | DILEU vs. PLA | (-0.94, 1.07) | 0.99 |
| | PLA | 49.0 ± 7.5 | 49.9 ± 7.1 | G × T | 0.89 | | DILEU vs. LEU | (-0.83, 1.22) | 0.88 |
| Intracellular Water (L) | LEU | 31.1 ± 4.0 | 31.6 ± 4.0 | G | 0.57 | 0.007 | LEU vs. PLA | (-0.77, 0.67) | 0.99 |
| | DILEU | 29.4 ± 3.4 | 30.0 ± 3.5 | T | <0.001 | | DILEU vs. PLA | (-0.65, 0.82) | 0.96 |
| | PLA | 31.1 ± 4.8 | 31.7 ± 4.7 | G × T | 0.91 | | DILEU vs. LEU | (-0.62, 0.89) | 0.90 |
| Extracellular Water (L) | LEU | 17.9 ± 2.2 | 18.1 ± 2.1 | G | 0.59 | 0.010 | LEU vs. PLA | (-0.39, 0.27) | 0.89 |
| | DILEU | 17.0 ± 1.8 | 17.3 ± 1.9 | T | <0.001 | | DILEU vs. PLA | (-0.34, 0.34) | 1.00 |
| | PLA | 18.0 ± 2.6 | 18.3 ± 2.5 | G × T | 0.87 | | DILEU vs. LEU | (-0.28, 0.41) | 0.89 |
| Vastus Lateralis (mm) | LEU | 173.6 ± 22.1 | 183.7 ± 24.4 | G | 0.16 | 0.024 | LEU vs. PLA | (-25.1, 28.6) | 0.99 |
| | DILEU | 175.0 ± 21.2 | 192.5 ± 24.3 | T | 0.12 | | DILEU vs. PLA | (-17.7, 36.0) | 0.68 |
| | PLA | 164.2 ± 22.7 | 172.5 ± 24.8 | G × T | 0.68 | | DILEU vs. LEU | (-20.1, 34.9) | 0.79 |
| Mid-Thigh (mm) | LEU | 351.7 ± 34.8 | 380.5 ± 51.0 | G | 0.03 | 0.045 | LEU vs. PLA | (-26.4, 61.8) | 0.59 |
| | DILEU | 326.4 ± 34.9 | 335.3 ± 26.6 | T | 0.04 | | DILEU vs. PLA | (-46.4, 41.8) | 0.99 |
| | PLA | 322.5 ± 46.7 | 333.6 ± 48.2 | G × T | 0.49 | | DILEU vs. LEU | (-65.1, 25.1) | 0.53 |

G × T = Interaction effect; T = Main effect for time; G = Main effect for group; p = probability level of making Type I error; 95% CI = 95% confidence intervals were computed on the observed changes from baseline between groups; Eta-squared ($\eta^2$) was used to estimate effect size where an $\eta^2$ of 0.01 or lower indicates a small, 0.06 indicates a medium effect and of 0.14 or larger indicates a large effect; kg = kilograms; DXA = dual energy x-ray absorptiometry; L = liters; mm = millimeter

amino acids [19]. This hypothesis is supported by the work of Paulussen et al. [20] who were seminal in demonstrating that ingestion of a LEU-LEU dipeptide may be more effective at stimulating an increase in MPS rates than free-LEU. The authors illustrated that greater plasma DILEU concentrations and greater DILEU area under the curve (AUC) were present when compared with the LEU group. Moreover, despite no differences in muscle protein breakdown rates, ingestion of 2 g DILEU increased plasma insulin concentrations, intramuscular LEU concentrations, and phosphorylated Akt similarly to an equal dose of LEU [20]. The ability for DILEU supplementation to elevate plasma DILEU concentrations and stimulate MPS to a greater degree than LEU raises intriguing questions about the efficacy of DILEU supplementation to promote resistance training adaptations compared to LEU supplementation.

**Table 5. Changes in performance variables.**

| Variable | Group | Week 0 | Week 10 | Mixed Factorial (p) | | ES ($\eta^2$) | Pairwise Comparisons | | |
|---|---|---|---|---|---|---|---|---|---|
| | | | | | | | | 95% CI | (p) |
| Bench Press 1RM (kg) | LEU | 104 ± 27 | 108 ± 28 | G | 0.46 | 0.112 | LEU vs. DILEU | (-10.7, 3.2) | 0.38 |
| | DILEU | 88 ± 23 | 98 ± 21 | T | <0.001 | | LEU vs. PLA | (-5.1, 8.7) | 0.80 |
| | PLA | 98 ± 25 | 105 ± 24 | G × T | 0.16 | | DILEU vs. PLA | (-1.5, 12.6) | 0.15 |
| Bench Press RTF (reps) | LEU | 8 ± 3 | 11 ± 3 | G | 0.29 | 0.138 | LEU vs. PLA | (-5.0, 2.2) | 0.60 |
| | DILEU | 9 ± 3 | 15 ± 6 | T | <0.001 | | DILEU vs. PLA | (-2.1, 5.1) | 0.56 |
| | PLA | 8 ± 2 | 12 ± 5 | G × T | 0.10 | | DILEU vs. LEU | (-0.73, 6.6) | 0.14 |
| Leg Press 1RM (kg) | LEU | 290 ± 67 | 335 ± 62 | G | 0.80 | 0.214 | LEU vs. PLA | (-10.6, 56.8) | 0.23 |
| | DILEU | 263 ± 75 | 324 ± 78 | T | <0.001 | | DILEU vs. PLA | (5.8, 73.2) | 0.02 |
| | PLA | 286 ± 74 | 307 ± 86 | G × T | 0.02 | | DILEU vs. LEU | (-18.0, 50.9) | 0.48 |
| Leg Press RTF (reps) | LEU | 13 ± 5 | 18 ± 5 | G | 0.08 | 0.203 | LEU vs. PLA | (-14.3, 5.0) | 0.47 |
| | DILEU | 13 ± 4 | 28 ± 8 | T | <0.001 | | DILEU vs. PLA | (-3.9, 15.5) | 0.32 |
| | PLA | 14 ± 8 | 24 ± 11 | G × T | 0.03 | | DILEU vs. LEU | (0.58, 20.3) | 0.04 |
| Total Strength (kg) | LEU | 395 ± 92 | 443 ± 87 | G | 0.72 | 0.219 | LEU vs. PLA | (-15.2, 53.9) | 0.36 |
| | DILEU | 351 ± 94 | 422 ± 94 | T | <0.001 | | DILEU vs. PLA | (6.8, 75.9) | 0.02 |
| | PLA | 383 ± 96 | 412 ± 108 | G × T | 0.02 | | DILEU vs. LEU | (-13.3, 57.3) | 0.29 |
| Total Reps (reps) | LEU | 21 ± 6 | 29 ± 7 | G | 0.04 | 0.249 | LEU vs. PLA | (-16.8, 4.7) | 0.36 |
| | DILEU | 22 ± 3 | 43 ± 12 | T | <0.001 | | DILEU vs. PLA | (-2.6, 19.0) | 0.16 |
| | PLA | 23 ± 8 | 36 ± 11 | G × T | 0.01 | | DILEU vs. LEU | (3.3, 25.3) | 0.009 |
| Relative Peak Power (W·kg⁻¹) | LEU | 10 ± 2.2 | 10.9 ± 2.4 | G | 0.71 | 0.136 | LEU vs. PLA | (—0.25, 2.59) | 0.12 |
| | DILEU | 11.3 ± 2.1 | 11.1 ± 2.3 | T | 0.96 | | DILEU vs. PLA | (-1.15, 1.62) | 0.91 |
| | PLA | 11.4 ± 1.5 | 10.9 ± 1.8 | G × T | 0.12 | | DILEU vs. LEU | (-2.36, 0.49) | 0.25 |
| Relative Mean Power (W·kg⁻¹) | LEU | 7.6 ± 1.3 | 7.7 ± 1.2 | G | 0.36 | 0.103 | LEU vs. PLA | (-0.43, 0.59) | 0.92 |
| | DILEU | 7.6 ± 1.0 | 7.4 ± 1.1 | T | 0.67 | | DILEU vs. PLA | (-0.77, 0.22) | 0.38 |
| | PLA | 8.1 ± 0.6 | 8.1 ± 0.7 | G × T | 0.21 | | DILEU vs. LEU | (-0.86, 0.15) | 0.21 |
| Total Work (J) | LEU | 18.6 ± 3.6 | 19.2 ± 3.3 | G | 0.67 | 0.076 | LEU vs. PLA | (-1.15, 1.71) | 0.88 |
| | DILEU | 18.0 ± 2.8 | 17.8 ± 2.6 | T | 0.30 | | DILEU vs. PLA | (-1.99, 0.87) | 0.60 |
| | PLA | 17.9 ± 2.5 | 18.2 ± 1.7 | G × T | 0.35 | | DILEU vs. LEU | (-2.27, 0.59) | 0.33 |
| Fatigue Index (%) | LEU | 48.3 ± 3.7 | 55.1 ± 4.0 | G | 0.52 | 0.144 | LEU vs. PLA | (-14.3, 11.7) | 1.00 |
| | DILEU | 47.3 ± 5.2 | 44.7 ± 5.7 | T | 0.43 | | DILEU vs. PLA | (-8.8, 23.0) | 1.00 |
| | PLA | 52.7 ± 53 | 53.4 ± 4.0 | G × T | 0.18 | | DILEU vs. LEU | (-23.0, 8.9) | 0.79 |
| Peak Propulsive Force (N) | LEU | 2050 ± 443 | 2096 ± 414 | G | 0.69 | 0.046 | LEU vs. PLA | (-122, 135) | 0.99 |
| | DILEU | 1956 ± 269 | 2045 ± 297 | T | 0.009 | | DILEU vs. PLA | (-73, 173) | 0.58 |
| | PLA | 1902 ± 421 | 1940 ± 426 | G × T | 0.54 | | DILEU vs. LEU | (-79, 166) | 0.66 |
| Jump Height (m) | LEU | 0.329 ± 0.07 | 0.324 ± 0.06 | G | 0.99 | 0.006 | LEU vs. PLA | (-0.035, 0.028) | 0.97 |
| | DILEU | 0.320 ± 0.11 | 0.320 ± 0.11 | T | 0.65 | | DILEU vs. PLA | (-0.028, 0.032) | 0.99 |
| | PLA | 0.324 ± 0.04 | 0.322 ± 0.06 | G × T | 0.92 | | DILEU vs. LEU | (-0.025, 0.035) | 0.91 |
| Isometric Mid-Thigh Pull (N) | LEU | 2038 ± 461 | 2227 ± 505 | G | 0.56 | 0.133 | LEU vs. PLA | (-675, 112) | 0.20 |
| | DILEU | 2123 ± 505 | 2303 ± 358 | T | <0.001 | | DILEU vs. PLA | (-674, 93) | 0.17 |
| | PLA | 2088 ± 440 | 2559 ± 377 | G × T | 0.13 | | DILEU vs. LEU | (-402, 384) | 0.99 |

G × T = Interaction effect; T = Main effect for time; G = Main effect for group; p = probability level of making Type I error; 95% CI = 95% confidence intervals were computed on the observed changes from baseline between groups; Eta-squared ($\eta^2$) was used to estimate effect size where an $\eta^2$ of 0.01 or lower indicates a small, 0.06 indicates a medium effect and of 0.14 or larger indicates a large effect; kg = kilograms; W = watts; J = joules; m = meters; N = newtons.

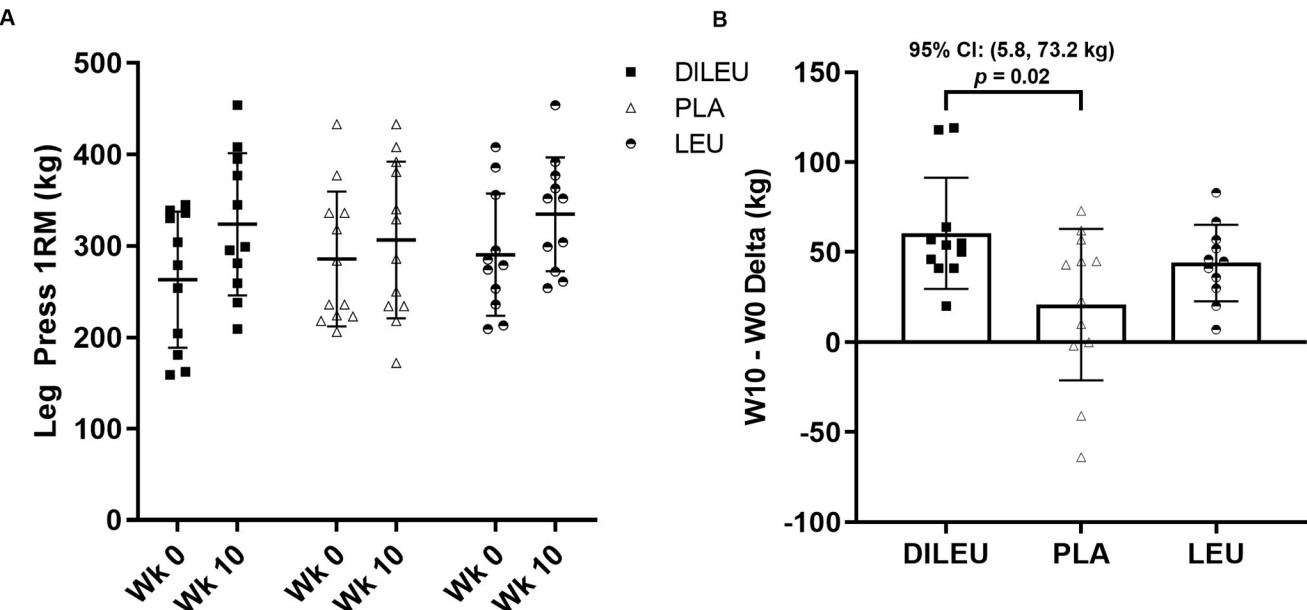

**Fig 2. (Sub-Panel a & b): Leg Press one-repetition maximum (1RM) in DILEU, LEU, and PLA supplemented groups.** Panel a: RAW data. Panel b: Delta.

Further to this point, ingestion of animal and plant protein increases DILEU levels in plasma [42], and ingested DILEU is in part being absorbed intact [20]. Even LEU ingestion has been shown to increase DILEU levels in plasma [20].

The preliminary findings of DILEU's impact on MPS and potential advantages over LEU supplementation led us to investigate its effectiveness in the context of longitudinal resistance-

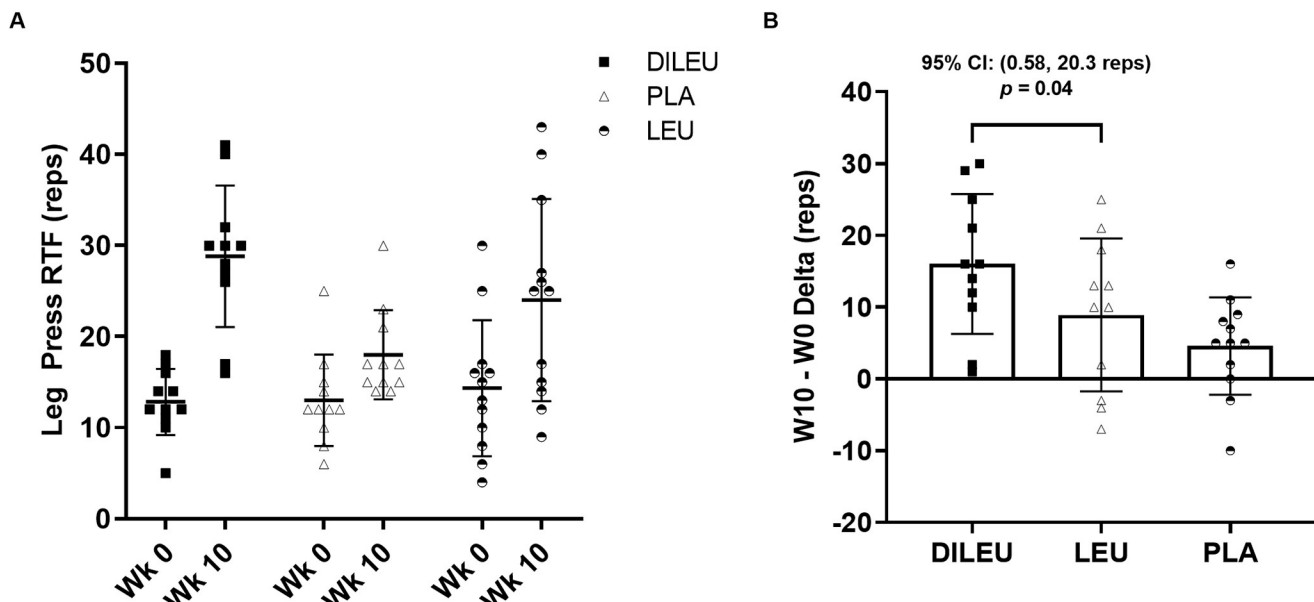

**Fig 3. (Sub-Panel a & b): Leg Press Repetitions to Failure (RTF) in DILEU, LEU, and PLA supplemented groups.** Panel a: RAW data. Panel b: Delta.

training adaptations. To ensure that any potential differences in muscle mass and strength between groups were not due to inherent differences in habitual diet, we evaluated nutritional intake throughout the study period. Additionally, to limit any training effects, participants were required to have been training for at least 12 months and possess a moderate level of relative baseline strength. While we observed no supplementation effect in upper-body strength or muscular endurance, our findings indicate that DILEU supplementation may lead to more substantial increases in muscular strength and muscular endurance compared to LEU supplementation. This presents a notable contrast within available research which has consistently showed no positive impact of LEU or essential amino acid supplementation on strength outcomes [43–45]. For instance, Spillane et al. [45] conducted an 8-week study in untrained participants engaged in a heavy resistance training program 4 days/week for 8 weeks and found that those consuming BCAA exhibited similar improvements in upper and lower body strength and endurance performance compared to those consuming PLA. Similarly, Mobley et al. [44] observed that untrained participants engaged in a 12-week whole body resistance training program 3 days/week exhibited comparable improvements in upper and lower body strength when supplementing with 3.0 g·d$^{-1}$ LEU compared to whey or soy protein (standardized to LEU content), and PLA. Additionally, Aguiar et al. [43] had untrained participants complete a 2 day/week hypertrophic resistance training program for 8 weeks and found no significant differences in lower leg strength when supplementing with 3.0 g·d$^{-1}$ LEU compared to PLA. The results of these studies and others [46–50] indicate that LEU supplementation combined with other essential amino acids does not result in greater strength gains than resistance training alone. However, these studies, except for that done by Ratamess et al. [49] and Kerksick et al. [48], differ significantly from ours in terms of participant training status, a factor that is likely to influence both the baseline MPS and the response to training and supplementation. Trained muscles exhibit distinct physiological characteristics compared to untrained muscles, including altered responsiveness to anabolic stimuli [4]. This could partly explain why DILEU supplementation was more effective, as resistance-trained muscles may utilize DILEU differently, with unique dipeptide structure of DILEU being a key aspect to consider. Dipeptides are known for their enhanced absorption rates compared to free amino acids [19], potentially leading to more efficient utilization for MPS in trained individuals, or the targeting of other mTORC1 regulatory events, such as mTORC1 translocation to the lysosome [20]. This enhanced efficacy could explain the more pronounced muscular strength and endurance gains observed in our study compared to those involving untrained individuals and different forms of amino acids.

Moving beyond these observations, our study failed to demonstrate any significant effects of supplementation on changes in fat-free mass and fat mass. This outcome is not surprising, considering prior investigations examining the impact of essential amino acid and LEU supplementation on hypertrophy following longitudinal resistance training [43–45, 48]. For instance, Aguiar et al [43] showed no supplementation effect on muscle mass or muscle thickness in untrained participants supplementing with 3 g LEU while completing an 8 week resistance training program. Spillane et al. [45] reported no significant changes in lean-body mass among untrained males ingesting 9 g·d$^{-1}$ of BCAA (4.5 g LEU) compared to a PLA group. Similarly, Mobley et al. [44] found no significant differences in total body muscle mass or VL muscle thickness among participants receiving 3 g·d$^{-1}$ of LEU compared to soy and whey protein, or PLA.

The lack of effects of LEU or essential amino acid supplementation on hypertrophy responses in our study and others [43, 44, 48] may be due to the fact that participants were already consuming adequate energy and protein. Participants in our study reported maintaining adequate energy and protein intake throughout the study, with daily energy intake ranging

from 31.0 to 33.7 kcal·kg$^{-1}$·d$^{-1}$ and protein intake ranging from 1.6 to 1.9 g·kg$^{-1}$·d$^{-1}$. This aligns with general protein intake recommendations [29, 51], which advocate for a protein intake of at least 1.2 g·kg$^{-1}$·d$^{-1}$ of protein to support muscle anabolism in conjunction with resistance training. The adherence of our participants to these nutritional guidelines suggests that they were likely in a state of nutritional adequacy, potentially diminishing the additional benefits that DILEU or LEU might offer. Furthermore, our findings underscore the multifaceted nature of muscle hypertrophy, which extends beyond the scope of supplementation. Factors such as energy and macronutrient intake [43, 52, 53], exercise volume [54], training age [55], and genetic predispositions [56] play significant roles in muscle development. Therefore, while LEU is integral to MPS, supplementation with DILEU or essential amino acids alone may not significantly enhance muscle hypertrophy response in populations already meeting their dietary requirements [18].

This study has potential limitations. Firstly, although the 10-week duration of the supplementation and resistance training program aligns with the typical duration in studies of this nature [23–25, 57], it is worth noting that a more extended investigation might have revealed more pronounced differences in adaptations between groups, potentially leading to statistically significant differences being identified between groups. Secondly, the number of subjects, while sufficient for detecting small between-group effects, remained relatively low for this study design. Consequently, we may not have possessed enough statistical power to detect small between-group outcomes. Furthermore, the absence of direct exercise training supervision may be viewed as a limitation, given previous findings indicating greater strength increases are observed when direct supervision occurs [58]. However, the direct impact of supervision on body composition changes remains uncertain. Nevertheless, while participants in the present study were not directly supervised for all workouts, they were required to submit daily photos after completing each workout and submit participant diary logs of their supplement consumption. Additionally, participants were required to routinely log their nutrition and frequently return to the lab to receive a new supply of their assigned supplements. In this respect, all participant interactions, whether during study visits, supervised workouts, supplement pick-up, or routine check-ins, were used to counsel participants on meeting their assigned nutritional goals and to review the appropriate progression of the loads they were using as part of their exercise program.

Key strengths of this investigation include its randomized, double-blind, placebo-controlled design. Additionally, we recruited resistance-trained participants who reported at least 12 months of resistance training experience to minimize the rapid increases in muscular strength and power as a result of neuromuscular adaptations that occur quickly after commencing a resistance training program [59]. Future research should explore the mechanisms through which DILEU improves lower-body strength and muscular endurance, in addition to DILEU's broader applications across different demographics, primarily aging populations.

## Conclusions

In conclusion, 10-weeks of supplementation with 2 g of DILEU resulted in significantly greater increases in maximal leg press strength and total strength (leg press + bench press) when compared to PLA. Additionally, DILEU ingestion was responsible for significantly greater increases in leg press and total repetitions (leg press + bench press) when compared to the changes observed in LEU.

## Supporting information

**S1 File.**
(PDF)

**S1 Checklist. CONSORT 2010 checklist of information to include when reporting a randomized trial[a].**
(PDF)

## Acknowledgments

The authors would like to extend our appreciation to all of the participants for their engagement in this study.

## Author Contributions

**Conceptualization:** Connor J. Gaige, Martin Purpura, Ralf Jager, Chad M. Kerksick.

**Formal analysis:** Chad M. Kerksick.

**Investigation:** Anthony M. Hagele, Joesi M. Krieger, Connor J. Gaige, Kevin F. Holley, Kristen N. Gross, Joshua M. Iannotti, Leah E. Allen, Paige J. Sutton, Logan S. Orr, Petey W. Mumford, Chad M. Kerksick.

**Methodology:** Kristen N. Gross, Martin Purpura, Ralf Jager, Chad M. Kerksick.

**Project administration:** Anthony M. Hagele.

**Visualization:** Anthony M. Hagele.

**Writing – original draft:** Anthony M. Hagele, Chad M. Kerksick.

**Writing – review & editing:** Anthony M. Hagele, Joesi M. Krieger, Connor J. Gaige, Kevin F. Holley, Kristen N. Gross, Joshua M. Iannotti, Leah E. Allen, Paige J. Sutton, Logan S. Orr, Petey W. Mumford, Martin Purpura, Ralf Jager, Chad M. Kerksick.

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
