## [Decision Letter · Decision Letter 0]

16 Jun 2024

PONE-D-24-07083Dileucine Ingestion, but not Leucine, Increases Lower Body Strength and Performance Following Resistance Training: A Double-Blind, Randomized, Placebo-Controlled TrialPLOS ONE

Dear Dr. Kerksick,

Thank you for submitting your manuscript to PLOS ONE. After careful consideration, we feel that it has merit but does not fully meet PLOS ONE’s publication criteria as it currently stands. Therefore, we invite you to submit a revised version of the manuscript that addresses the points raised during the review process.

We look forward to receiving your revised manuscript.

Kind regards,

Krzysztof Durkalec-Michalski, Ph.D

Academic Editor

PLOS ONE

https://doi.org/10.1186/s12970-020-00394-1

https://doi.org/10.1016/j.isci.2023.108643

In your revision ensure you cite all your sources (including your own works), and quote or rephrase any duplicated text outside the methods section. Further consideration is dependent on these concerns being addressed.

 [Funding was acquired by CMK. Grant #: 01-2020. This study was funded by Ingenious Ingredients, LLC (https://ing2.com/). Martin Purpura and Ralf Jager are principals of Ingenious Ingredients, LLC,  the sponsor of the study, and  were involved in conceptualizing and designing the study, and reviewing and editing the manuscript but were not involved in data collection or data analysis.].  

[I have read the journal's policy and the authors of this manuscript have the following competing interests: M.P and R.J are principals of Ingenious Ingredients, LLC, the sponsor of the study, and inventors of numerous patent applications for the use of dileucine but have not been involved in data collection or analysis. All other authors declare no competing interests.]. 

6. We note that your Data Availability Statement is currently as follows: [All relevant data are within the manuscript and its Supporting Information Files.]

Reviewers' comments:

Reviewer's Responses to Questions

**Comments to the Author**

1. Is the manuscript technically sound, and do the data support the conclusions?

Reviewer #1: Yes

Reviewer #2: Yes

Reviewer #3: Yes

2. Has the statistical analysis been performed appropriately and rigorously? 

Reviewer #1: Yes

Reviewer #2: Yes

Reviewer #3: Yes

3. Have the authors made all data underlying the findings in their manuscript fully available?

Reviewer #1: Yes

Reviewer #2: Yes

Reviewer #3: Yes

4. Is the manuscript presented in an intelligible fashion and written in standard English?

Reviewer #1: Yes

Reviewer #2: Yes

Reviewer #3: Yes

5. Review Comments to the Author

Reviewer #1: A randomized controlled clinical trial was conducted which aimed to assess the effects of DILEU supplementation on resistance training adaptations. DILEU supplementation enhanced lower body strength and muscular endurance in resistance-trained males more effectively than LEU or PLA.

Minor revisions:

1- Line 100: State the statistical testing method that achieves 80% power.

2- If the interaction effect is significant, provide an interpretation of the results, but do not test main effects because the tests for main effects are uninteresting in light of significant interactions. If interaction effects are non-significant, drop the interaction effects from the model and test the main effects. Determining which results to present when testing interactions is often a multi-step process.

Reviewer #2: line 217: workload greater than 7.5% often produce greater Wingate results. Why did you choose 7.5%?

line 222: ??? Participants ramped up rpm following when resistance applied?. Typically, rpm are maximized with no load and once maximized the workload is engaged. Your approach would likely generate less than maximal values.

line 226: What about the fatigue index?

line 263: Often supplements do not contain the quantity of compounds claimed or contain others not listed on the label. Please indicate the analyzed supplement contents or list as a limitation.

line 295: ....endpoints of ....

TABLE 2. Weight should be mass

Relative bench of leg press ??? you need to xplain what this is and add units

line 318: Seems to be be methods not results

line 334: No change in muscle thickness with training? Does this indicate the training stimulus was inadequate? Please discuss.

line 372. Why only males?

line 441 and elsewhere. kcal/kg/d is mathematically incorrect. Should be kcal . kg-1 . d -1 (dots should be raised to the centre of line and -1 written as superscripts) Sorry the review pane doesn't allow me to write correctly.

Figures: Use open and closed symbols as well as differing shapes symbols as this makes results more clear.

Reviewer #3: [1] P3 L45: If the outcome of the study is performance, the authors should not mention health and ageing in the introduction. It is also important to explain how MPS could affect performance.

[2] P3 L56:  Please indicate some sources of protein with higher leucine content.

[3] The third paragraph is longer than expected, and the mechanisms are very detailed in the introduction section. Please summarize the main ideas.

[4] P4 L90: Why do the authors choose ten weeks? This should be explained in the introduction – the chronic effects of amino acids – using relevant literature.

[5] P5 L94: Why do the authors randomized based on fat-free mass when the main outcomes were performance?

[6] P5 L104: The sentence about body composition needs to be clarified. Please rephrase.

[7] P11 L249: Why do the authors use the Harris-Benedict equation?

[8] P14 Table 2: Please add the t and p values to the table.

[9] P17 Table 4: Please adjust Table 4. It is not presentable.

[10] P17 L39: Please summarize the direction of the differences.  

[11] The discussion is well written. Congratulations.

6. PLOS authors have the option to publish the peer review history of their article (what does this mean?). If published, this will include your full peer review and any attached files.

Reviewer #1: No

Reviewer #2: **Yes: **Peter WR Lemon

Reviewer #3: No

---

## [Author Response · Author response to Decision Letter 0]

12 Jul 2024

July 11, 2024

RE: PONE-D-24-07083

PLOS One Editorial Board:

On behalf of the authors, we would like to re-submit the following manuscript, “Dileucine Ingestion, but not Leucine, Increases Lower Body Strength and Performance Following Resistance Training: A Double-Blind, Randomized, Placebo-Controlled Trial” to PLOS One for consideration to be published. We have addressed all comments brought forth by the editor in their email dated June 16, 2024 as well as all of the reviewer comments.

We have provided our responses to the editor immediate below and following that we have provided a point-by-point response to all of the comments brought forth by the reviewers.

We look forward to hearing any further comments on our paper. Thank you.

Sincerely,

All comments have been addressed by our authors and our responses are included in our revision.

Please let me know if you need any further information.

Chad M. Kerksick, PhD

Assistant Dean, Research & Innovation

Director, Exercise and Performance Nutrition Laboratory

Lindenwood University

(636) 627-4629

ckerksick@lindenwood.edu

Editor Comments

1) PLOS Formatting and Style Requirements.

RESPONSE: We have done our best to align our submitted paper with our interpretation and understanding of the formatting guidelines for PLOS One.

2) Overlapping Text

RESPONSE: We have amended version of our current text. The overlapping text consists primarily of areas within our methods where we have explained our testing procedures. Minimal to no overlap should be found in the abstract, introduction, results, discussion, figures, and tables.

3) Grant Information

RESPONSE: My university does not issue official grant numbers as all of our grants are organized by the title of the project. The funding information in our submission is correct:

Recipient: Chad M. Kerksick

Award Number: None

Sponsor: Ingenious Ingredients, L.P.

4) Financial Disclosure

RESPONSE: Here is an updated statement highlighting that much of what was being asked was already provided.

Funding was acquired by CMK. This study was funded by Ingenious Ingredients, LLC (https://ing2.com/). Martin Purpura and Ralf Jager are principals of Ingenious Ingredients, LLC, the sponsor of the study. The funders assisted in conceptualizing and designing the study, and reviewing and editing the manuscript. The funders were not involved in data collection, data analysis, or data interpretation. Additionally, the sponsors played no role in the decision to publish, prepare, or revising the manuscript.

5) Competing Interests

RESPONSE: This does not change our statement. Here is the revised statement with the added sentence requested.

I have read the journal's policy and the authors of this manuscript have the following competing interests: M.P and R.J are principals of Ingenious Ingredients, LLC, the sponsor of the study, and inventors of numerous patent applications for the use of dileucine but have not been involved in data collection or analysis. All other authors declare no competing interests. This does not alter our adherence to PLOS ONE policies on sharing data and materials as there are no restrictions on sharing data and/or materials.

6) Data Availability

RESPONSE: Our submission contains a file that has all of the data used for this manuscript.

Review Comments to the Author

Reviewer #1: A randomized controlled clinical trial was conducted which aimed to assess the effects of DILEU supplementation on resistance training adaptations. DILEU supplementation enhanced lower body strength and muscular endurance in resistance-trained males more effectively than LEU or PLA.

Minor revisions:

1- Line 100: State the statistical testing method that achieves 80% power.

Author Response: Thank you for your comment. We have revised the manuscript to include the specific statistical testing method used to achieve 80% power.

2- If the interaction effect is significant, provide an interpretation of the results, but do not test main effects because the tests for main effects are uninteresting in light of significant interactions. If interaction effects are non-significant, drop the interaction effects from the model and test the main effects. Determining which results to present when testing interactions is often a multi-step process.

Author Response: We have revised the results section to follow your recommendations. Specifically, we have focused on interpreting significant interaction effects and dropped non-significant interaction effects from the model, testing only the main effects in those cases.

Reviewer #2: line 217: workload greater than 7.5% often produce greater Wingate results. Why did you choose 7.5%?

Author Response: Thank you for your comments. We chose a workload of 7.5% of body weight based on standard testing procedures and this workload being used as the initial workload with Wingate testing (Inbar et al. 1996). In this respect, we also recognize the work of others (Pazin et al. EJAP 2011 and Silveira-Rodrigues et al. Fatigue Biomed Hlth Behav 2021) who have demonstrated that a workload of 7.5% may not align with peak power production and a different workload prescription may have resulted in different power and work production numbers.

While this work is valuable, our research design was intended to evaluate the changes in our measured endpoints across a specified period of time. Thus, we were more interested in being able to administer a similar dose of testing stress at each testing point as opposed to being able to ensure we were using the best protocol to achieve peak power production. 

line 222: ??? Participants ramped up rpm following when resistance applied?. Typically, rpm are maximized with no load and once maximized the workload is engaged. Your approach would likely generate less than maximal values.

Author Response: Thank you for pointing out our error in explaining the methodology used for the Wingate test. Our protocol is consistent with standard testing where participants began pedaling at maximal RPM with no load and once maximal RPM was achieved, the resistance was applied. We have revised the manuscript to reflect this more clearly.

line 226: What about the fatigue index?

Author Response: We have included the calculation and analysis of the fatigue index in the revised manuscript.

line 263: Often supplements do not contain the quantity of compounds claimed or contain others not listed on the label. Please indicate the analyzed supplement contents or list as a limitation.

Author Response: We have included a certificate of analysis performed on the same lot of product used in our clinical trial. We have also indicated that the content was verified.

line 295: ....endpoints of ....

Author Response: Thank you. We have amended this sentence.

TABLE 2. Weight should be mass

Author Response: We have revised Table 2 to use the term "body mass" instead of "weight" to accurately reflect the measurement. The revised table header now reads "Body Mass"

Relative bench of leg press ??? you need to xplain what this is and add units

Author Response: We apologize for the lack of clarity. The terms “Relative Leg Press” and “Relative Bench Press” refers to the relative strength calculated for both the leg press and bench press. Specifically, we calculated the relative strength by dividing the one-repetition maximum (1RM) for each exercise by the participant’s body mass. This value is expressed as a ratio (e.g., kg/kg). We have revised the manuscript to include the appropriate units.

line 318: Seems to be be methods not results

Author Response: According to the CONSORT checklist, the flow of participants including the number of participants who were recruited, consented, and completed the study, should be reported in the results section. Therefore, we have included this information in the results section to align with these guidelines.

line 334: No change in muscle thickness with training? Does this indicate the training stimulus was inadequate? Please discuss.

Author Response: Our apologies as we incorrectly reported our changes in ultrasound muscle thickness as we observed a significant main effect for mid-thigh muscle thickness to increase (p = 0.04) while vastus lateralis (p = 0.12) did not quite reach statistical significance.

The lack of a significant main effect for vastus lateralis (p = 0.12) muscle thickness could have been due to differences in the measurement approaches that we employed in our study as well as subtle differences in each measurements, although distinct efforts were made to maximize our reliability with our muscle thickness measures. Certainly, lack of intensity or volume cannot be entirely ruled out within the given timeframe to instigate muscle hypertrophy as well as exercise selection and nutritional factors.

Other key considerations that impact this outcomes were that all participants were resistance-trained for at least 12 months, which might have influenced the outcomes. Advanced trainees often require a higher stimulus to achieve further hypertrophy compared to novice trainees. While muscle thickness and fat-free mass did not significantly change, strength and muscular endurance did increase in all groups. This suggests that the training program was effective at increasing neuromuscular adaptations and strength, but not at inducing significant hypertrophy. It is possible that the duration of the study was not long enough to observe hypertrophic responses in a population that is already well-trained. These limitations have been mentioned in the manuscript.

line 372. Why only males?

Author Response: As this was the first human clinical study being completed using dileucine in a longitudinal study design to evaluate its potential to augment exercise training adaptations, the rationale for including only male participants in this study was to examine resistance training adaptations and supplementation effects in a more homogeneous population, which helps to control for confounding variables. Including only males allowed us to reduce variability and increase the internal validity of the study. With reduced variability our ability to identify any treatment effects should have been bolstered. We acknowledge the importance of including female participants in future studies to determine if the findings are generalizable across sexes. Future research should aim to explore these effects in female populations.

line 441 and elsewhere. kcal/kg/d is mathematically incorrect. Should be kcal . kg-1 . d -1 (dots should be raised to the centre of line and -1 written as superscripts) Sorry the review pane doesn't allow me to write correctly.

Author Response: We have corrected the notation throughout the manuscript to reflect the correct mathematical expression. 

Figures: Use open and closed symbols as well as differing shapes symbols as this makes results more clear.

Author Response: Thank you for your suggestion. We have revised the figures to use open and closed symbols as well as differing shapes to enhance the clarity of the results.

Reviewer #3: [1] P3 L45: If the outcome of the study is performance, the authors should not mention health and ageing in the introduction. It is also important to explain how MPS could affect performance.

Author Response: Thank you for your comments. We have revised the introduction to focus more on performance and provide a clear explanation of how MPS can affect performance outcomes.

[2] P3 L56: Please indicate some sources of protein with higher leucine content.

Author Response: We have amended this section to indicate the leucine content of various protein sources.

[3] The third paragraph is longer than expected, and the mechanisms are very detailed in the introduction section. Please summarize the main ideas.

Author Response: Thank you for your comment. We have revised this paragraph to summarize the main ideas more concisely and reduce the length.

[4] P4 L90: Why do the authors choose ten weeks? This should be explained in the introduction – the chronic effects of amino acids – using relevant literature.

Author Response: We chose a ten-week duration for the study to allow sufficient time to observe the chronic effects of amino acid supplementation on resistance training adaptations. We have included an explanation in the introduction to justify the choice of a 10-week duration for the study.

[5] P5 L94: Why do the authors randomized based on fat-free mass when the main outcomes were performance?

Author Response: We randomized participants based on fat-free mass to ensure balanced groups with similar muscle mass. Our primary endpoints were fat-free mass and leg press 1RM. By controlling for fat-free mass, we aimed to minimize variability in muscle mass between groups, ensuring that any observed differences in these primary outcomes were more likely attributable to the interventions rather than differences in baseline muscle mass.

[6] P5 L104: The sentence about body composition needs to be clarified. Please rephrase.

Author Response: We believe the confusion may be due to the term “lean mass”, which should be specified as “dry lean mass.” We have revised the sentence appropriately.

[7] P11 L249: Why do the authors use the Harris-Benedict equation?

Author Response: We used the Harris-Benedict equation in conjunction with the Mifflin-St. Jeor formula to provide a more comprehensive estimation of resting energy expenditure (REE) for participants. Both equations are well-established and widely used in clinical and research settings for estimating caloric needs. By averaging the results from both the Harris-Benedict and Mifflin-St. Jeor formulas, we aimed to account for potential variations in individual metabolic rates and provide a more robust estimation of energy requirements. Additionally, we have successfully used this approach in past studies to provide estimates of REE for participants.

Further, the utilization of these questions was simply to provide an estimate or target of where energetic needs were for our people. Both equations are well-validated for this type of use.

[8] P14 Table 2: Please add the t and p values to the table.

Author Response: Thank you for your comment. According to the current CONSORT guidelines, it is recommended to present baseline characteristics in a way that emphasizes the balance between groups rather than statistical comparisons (such as t and p values). The focus is on the description of the baseline characteristics to demonstrate the similarity of the groups at the start of the trial. Including statistical comparisons of baseline characteristics could lead to the incorrect interpretation that randomization was unsuccessful if significant differences are found by chance. Therefore, we have chosen not to include t and p values in Table 2 to align with the CONSORT guidelines recommended for use by PLoS One. We hope this explanation clarifies our decision. We believe this approach adheres to best practices for reporting randomized controlled trials and ensures the focus remains on the comparability of the groups rather than statistical significance of baseline differences. https://pubmed.ncbi.nlm.nih.gov/25616598/

[9] P17 Table 4: Please adjust Table 4. It is not presentable.

Author Response: We have adjusted the page orientation to enhance its presentation.

[10] P17 L39: Please summarize the direction of the differences. 

Author Response: We have amended this line and other areas to summarize the direction of the differences.

[11] The discussion is well written. Congratulations.

Author Response: Thank you!

---

## [Decision Letter · Decision Letter 1]

27 Sep 2024

PONE-D-24-07083R1Dileucine Ingestion, but not Leucine, Increases Lower Body Strength and Performance Following Resistance Training: A Double-Blind, Randomized, Placebo-Controlled Trial

PLOS ONE

 Dear Dr. Kerksick,

After careful consideration, we feel that it has merit but does not fully meet PLOS ONE’s publication criteria as it currently stands. Therefore, we invite you to submit a revised version of the manuscript that addresses the points raised during the review process.

COMMENTS:

In the work, all unit entries (text, tables) have still not been corrected in accordance with the reviewer's comments - they should be unified everywhere, e.g. "kcal∙kg^-1^" instead of "kcal/kg"; "W∙kg^-1^" instead of "W/kg”, „g∙kg^-1^∙day^-1^” instead of „g/kg/day”, „kg∙m^-2^” „kg/m^2^” etc.

Line 60 - insert a space between (6-8%) and [14,15].

line 70: first use - "DiLEU (DILEU)", - use the full name and abbreviations can be used in the rest of the text.

Line 258 – change „caloric” to „energy”.

Table 2 - The lack of statistical values makes it impossible to assess whether the participans did not differ "at the entrance" to the individual groups (LEU vs. DILEU vs. PLA). This point should be addedd so as not to leave readers with potential doubts.

Table 2 - In the unit description for Relative Leg/Bench Press, put "Body Mass" in the subscript.

Table 3 – insert „intake” after „Relative Energy”.

Table 4 – insert „mass” after "Dry lean".

In the revised description of the results, there are probably mistakes in some points - these descriptions do not match the data from the tables.

- In the line 359 - the authors write "a significant increase in bench press 1RM for DILEU compared to PLA (p = 0.02; 95% CI: 6.8, 75.9 kg)" - here it should rather be "total strength" (see table 5).

- in the 374 - similarly as above. Authors write „…increase in leg press RFT” - and it should be " …increase in total reps..".

- The whole manuscript should be checked in this respect to make sure all descriptions are correct.

Lines 383 and 384 - p values are incorrect and there is an error in the descriptions („p = 0.0.54”;  „p = 0.0.65” and „p = 0.0.92”). This should be corrected and the entire text/tables/figures should be checked again.

Line – „calories” are not consumed - it is de facto a unit. Please correct the sentence and use "energy intake".==============================

We look forward to receiving your revised manuscript.

Kind regards,

Krzysztof Durkalec-Michalski, Ph.D

Academic Editor

PLOS ONE

Journal Requirements:

Reviewers' comments:

Reviewer's Responses to Questions

**Comments to the Author**

1. If the authors have adequately addressed your comments raised in a previous round of review and you feel that this manuscript is now acceptable for publication, you may indicate that here to bypass the “Comments to the Author” section, enter your conflict of interest statement in the “Confidential to Editor” section, and submit your "Accept" recommendation.

Reviewer #1: All comments have been addressed

Reviewer #2: All comments have been addressed

2. Is the manuscript technically sound, and do the data support the conclusions?

Reviewer #1: (No Response)

Reviewer #2: Yes

3. Has the statistical analysis been performed appropriately and rigorously? 

Reviewer #1: (No Response)

Reviewer #2: Yes

4. Have the authors made all data underlying the findings in their manuscript fully available?

Reviewer #1: (No Response)

Reviewer #2: Yes

5. Is the manuscript presented in an intelligible fashion and written in standard English?

Reviewer #1: (No Response)

Reviewer #2: Yes

6. Review Comments to the Author

Reviewer #1: (No Response)

Reviewer #2: There are still several places where the units are mathematically incorrect - see line 113 and elsewhere!

7. PLOS authors have the option to publish the peer review history of their article (what does this mean?). If published, this will include your full peer review and any attached files.

Reviewer #1: No

Reviewer #2: **Yes: **Pete Lemon

---

## [Author Response · Author response to Decision Letter 1]

4 Oct 2024

COMMENTS:

1. In the work, all unit entries (text, tables) have still not been corrected in accordance with the reviewer's comments - they should be unified everywhere, e.g. "kcal∙kg-1" instead of "kcal/kg"; "W∙kg-1" instead of "W/kg”, „g∙kg-1∙day-1” instead of „g/kg/day”, „kg∙m-2” „kg/m2” etc.

Author Response: Thank you for your feedback. We have now carefully revised and unified all unit entries throughout the manuscript, ensuring consistency across both the text and tables. All units have been updated to the correct scientific notation per your suggestion.

2. Line 60 - insert a space between (6-8%) and [14,15].

 Author Response: Thank you for pointing this out. We have inserted the space between "(6-8%)" and the citation "[14,15]" as requested.

3. line 70: first use - "DiLEU (DILEU)", - use the full name and abbreviations can be used in the rest of the text.

 Author Response: Thank you for your suggestion. We have now provided the full name at its first mention followed by the abbreviation in parentheses, and used the abbreviation throughout the remainder of the text.

4. Line 258 – change „caloric” to „energy”.

 Author Response: Thank you for your suggestion. We have replaced “caloric” with “energy” throughout the manuscript as requested.

5. Table 2 - The lack of statistical values makes it impossible to assess whether the participans did not differ "at the entrance" to the individual groups (LEU vs. DILEU vs. PLA). This point should be addedd so as not to leave readers with potential doubts.

Author Response: Thank you for your observation. We have now added the relevant statistical values to Table 2 to clarify that there were no significant differences between the groups at baseline.

6. Table 2 - In the unit description for Relative Leg/Bench Press, put "Body Mass" in the subscript.

Author Response: Thank you for the comment. We have revised the unit description in Table 2 to include “Body Mass” in the subscript as requested.

7. Table 3 – insert „intake” after „Relative Energy”.

Author Response: We have inserted “intake” after “Relative Energy” in Table 3

8. Table 4 – insert „mass” after "Dry lean".

Author Response: We have inserted “mass” after “Dry lean” in Table 4

9. In the revised description of the results, there are probably mistakes in some points - these descriptions do not match the data from the tables.

- In the line 359 - the authors write "a significant increase in bench press 1RM for DILEU compared to PLA (p = 0.02; 95% CI: 6.8, 75.9 kg)" - here it should rather be "total strength" (see table 5).

- in the 374 - similarly as above. Authors write „…increase in leg press RFT” - and it should be " …increase in total reps..".

- The whole manuscript should be checked in this respect to make sure all descriptions are correct.

 Author Response: Thank you for identifying these discrepancies. We have corrected the descriptions in the highlighted lines and thoroughly reviewed the entire manuscript to ensure that all results descriptions are accurate and match the data in the tables.

10. Lines 383 and 384 - p values are incorrect and there is an error in the descriptions („p = 0.0.54”; „p = 0.0.65” and „p = 0.0.92”). This should be corrected and the entire text/tables/figures should be checked again.

 Author Response: Thank you for pointing this out. We have corrected the p-values in lines 383 and 383 and reviewed the entire manuscript, including text, tables, and figures, to ensure all p-values are accurate and correctly reported.

11. Line – „calories” are not consumed - it is de facto a unit. Please correct the sentence and use "energy intake".

Author Response: Thank you for the suggestion. We have corrected the sentence to use “energy intake” instead of calories to accurately reflect the concept.

---

## [Editor Report · Decision Letter 2]

17 Oct 2024

Dileucine Ingestion, but not Leucine, Increases Lower Body Strength and Performance Following Resistance Training: A Double-Blind, Randomized, Placebo-Controlled Trial

PONE-D-24-07083R2

Dear Dr. Kerksick,

We’re pleased to inform you that your manuscript has been judged scientifically suitable for publication and will be formally accepted for publication once it meets all outstanding technical requirements.

Kind regards,

Krzysztof Durkalec-Michalski, Ph.D

Academic Editor

PLOS ONE

---

## [Editor Report · Acceptance letter]

28 Oct 2024

PONE-D-24-07083R2 

PLOS ONE

Dear Dr. Kerksick, 

I'm pleased to inform you that your manuscript has been deemed suitable for publication in PLOS ONE. Congratulations! Your manuscript is now being handed over to our production team.

Kind regards, 

on behalf of

Dr. Krzysztof Durkalec-Michalski 

Academic Editor

PLOS ONE